# Evaluation of soil quality in Inner Mongolia desert steppe—A case study of Siziwang Banner

Hongtao Jiang[1,2,3☯], Xiaojia Li[1,2,3☯], Chunrong Guo[1,4☯]*, Yufeng Zhang[1,2]*, Ruiping Zhou[1,2], Chunxing Hai[1], Zheru Hu[5]

**1** College of Geographical Science, Inner Mongolia Normal University, Hohhot, Inner Mongolia, China, **2** Inner Mongolia Autonomous Region Land Use and Renovation Engineering Technology Research Center, Hohhot, Inner Mongolia, China, **3** Laboratory of Mongolian Plateau Environment & Global Change, Hohhot, Inner Mongolia, China, **4** Inner Mongolia Land Use and Regulation Engineering Technology Research Center, Hohhot, Inner Mongolia, China, **5** School of Public Management, Inner Mongolia University, Hohhot, Inner Mongolia, China

☯ These authors contributed equally to this work.
* guochunrong2022@163.com (CG); zhangyufeng@imnu.edu.cn (YZ)

## Abstract

As a natural ecological fragile region, the vast desert steppe in the Inner Mongolia has a developed animal husbandry, and thus posed great impacts on soil quality. In order to accurately evaluate the current situation of soil quality in the desert steppe, it is therefore imperative to adopt a suitable method to effectively assess the soil quality in the region. In this study, the minimum data set (MDS) was established with the help of principal component analysis, Norm value calculation, and correlation analysis, and four indicators, including organic matter, sand grains, soil erosion degree, and pH, were established to evaluate the soil quality of the desert steppe in the Siziwang Banner, a county in the Inner Mongolia. The results from the minimum data set (MDS) method were validated based on the total data set (TDS) method, and the validation indicated that the MDS method can be representative of the soil quality of the study area. The results indicated: 1) the soil quality index (SQI) of 0–30 cm in more than 90% of the study area falls in the range of 0.4 and 0.6 (medium level), while the better level (SQI ≥0.6) only accounted less than 10% of the study area; 2) For the MDS indexes, soil organic matter content at all depths decreased in the southern mountains, central hills, and northern plateau, which is consistent with the changing trends of SQI; 3) The sand grain was the dominant particle in the study region, which was in accordance with the intense wind erosion; 4) The negative correlation was found between the soil pH value and SQI (the high value in pH corresponded to the low value in SQI), which reflected that soil pH has a more stressful effect on the local vegetation. Overall, the MDS indexes in this study can objectively and practically reflect the soil quality in the study area, which can provide a cost effective method for SQI assessment in the desert steppe, which is important for the further grassland ecological construction and grassland management to improve the soil quality in the desert steppes.

**Data Availability Statement:** All relevant data are within the manuscript.

**Funding:** This study was supported by the Key Project of Inner Mongolia Autonomous Region Philosophy and Social Science Planning (Research on Optimization of Industrial Structure Based on

Natural Resources and Conditions in Inner Mongolia, 2022NDA219) (Yufeng Zhang), and the funder participated in the program involvement of the project and funded the publication cost of the results. This study was also supported by Major Special Projects of Science and Technology Plan of Inner Mongolia Autonomous Region (ZDZX2018058) (Ruiping Zhou), which funded the field sampling and soil sample testing costs.

**Competing interests:** The authors have declared that no competing interests exist.

# 1 Introduction

Desert steppe in the Mongolian plateau is an important part of the arid region across the entire world, with the characteristics of sparse vegetation over, intense soil erosion, and serious soil degradation. The area of desert steppe in the Inner Mongolia Autonomous Region, China, is about 112,000 km$^2$, accounting for more than 10 percent of the total area of Inner Mongolia [1]. Due to years of continuous overgrazing, population growth and extreme arid climate, desert grasslands are often considered as "ecologically fragile areas" [1], which leading to serious wind erosion on desert grasslands and exacerbating the process of desertification [2], resulting in the poor soil quality. In this context, an accurate and comprehensive assessment of soil quality in the desert steppe region is of great importance in order to fully understand its current situation and to further protect and remediate the soil.

Currently soil quality assessment system frameworks are widely used in many countries/areas of the world, such as the Comprehensive Soil Health Assessment in the United States [3,4], CASH widish agricultural soil quality evaluation system [5], and the forest soil quality Regression equation in Alicante, Spain [6]. Based on these assessment systems and methods, current studies mainly focused on the quantitative assessments of the effects of management measures on soil quality based on the soil quality index (SQI) [7–10]. According to the literature review [11], 415 Chinese and English literature articles on soil quality assessment and soil health assessment have been found since 1990, 155 of which used the Minimum Data Set (MDS) method for soil quality assessment. By using the fewest indicators to better monitor and reflect soil quality in the region, this method has been widely used in soil quality assessment of cultivated land [12–16], forest land [17–19] and artificial grasslands [20,21]. It can also be used to assess the effects of land use change, fertilization, changes in planting system, etc., on soil quality change [16,22–26]. Based on the literature review, it can be concluded that currently a large number of studies regarding soil quality assessments are conducted mainly on cultivated land and agricultural systems while studies on the assessments of soil quality in the desert steppe are relatively rare. In addition, there is little research to analyze and select the most optimal assessment methods and key indicators.

Current researches on desert steppe mainly focused on land conservation and ecological restoration, and assessed the impact of different types of land use, different grazing intensities and fence treatments, and soil and water conservation measures on soil quality. [27–33]. These studies are valuable for preventing soil erosion, improving vegetation coverage and combating desertification. However, since most of these studies did not address changes in soil quality, it is difficult to fully understand soil quality in desert steppe based on these studies. In addition, these studies are often conducted on relatively small sample sizes, which prevents us from fully understanding large-scale soil quality. Therefore, it is urgent to carry out large-scale studies that focused on soil quality assessments in the desert steppe.

Located in the middle part of Inner Mongolia, the Siziwang Banner is a total animal husbandry county in the desert steppe in the Mongolia Plateau with a total area of 25513km$^2$. The Siziwang Banner borders Mongolia and represents the desert steppe of Mongolia and China. Moreover, the Siziwang Banner is also an important northern sand belt area in the national project of "two screens and three belts" [33]. Given the severe ecological situation faced by the desert steppe in northern China, it is necessary to choose appropriate methods indicators. It is of great significance for the restoration and improvement of soil environmental quality in desert steppe areas to carry out the evaluation of soil environmental quality in desert steppe and establish the method of selecting key indicators for the evaluation of soil environmental quality in desert steppe.

## 2 Site description

The Siziwang Banner is located in the desert steppe in the Mongolia Plateau with the range of 110˚ 19' 53"—112˚ 59' 37" E and 41˚ 11' 32"—43˚ 22' 31"N. The soil type is mainly chestnut soil. The landform in the Siziwang Banner was mainly consisted of mountains, hills and plateau, among which, the Yinshan Mountain was in the south part of the Siziwang Banner with the altitude of 1600–2100 m, the hill was in the central part at the average altitude of about 1400 m while the elavation of the flat open plateau in the north was about 1000–1300 m. The long-term average annual precipitation was about 100–300 m, which gradually decreased from south to north, and the annual precipitation was about 2400 m. The strong winds are frequent in the Siziwang Banner and the average daily wind speed for 68% of the days per year was higher than 6 m s$^{-1}$. The vegetation is mainly short grass with a low vegetation cover of about 9–20%. The harsh natural environment, together with the human activities, mainly animal husbandry, has made the Siziwang Banner suffer from intense soil erosion.

## 3 Materials and methods

### 3.1 Sample collection

The Siziwang Banner was divided into the 10×10 km grid, and the center of each grid was set as the sampling sites, however, several sampling sites can't be reached due to the lack of road, thus the locations of these sampling sites has been slightly adjusted. Finally, there were in total 38 sampling points in the Siziwang Banner, among which, 11 sites were in the south mountains, 14 sites in the central hills and 13 sites in the northern plain, respectively. Soil sampling was conducted in July 2019 and July 2020, for each sampling site, a small square quadrat with the side length of 1 m was set up at the center of the site. The four vertices and the center of the square were considered as the five sampling points for each site, for each point, the soil samples were collected at the depth of 0–5 cm, 5–10 cm, 10–15 cm, 15–20 cm, 20–25 cm, and 25–30 cm, respectively, then the collected soil samples of the same depth at the five points were uniformly mixed to increase the representativeness of the soils at the sampling sites. For each site, two parallel samples were collected nearby the sampling point within the area range of 100 m$^2$, then finally a total of 608 samples (38 sampling sites × 6 depths × 3) were collected. study area and sampling points were shown in (Fig 1).

### 3.2 Soil processing and analysis

The soil samples have been dried and sieved for the experiments of the physico-chemical properties. For all the properties, the classical experimental methods were chosen, such as, the soil pH is measured by the PXJ-1C precision millivolt-pH-ion activity meter method; organic matter content is measured by potassium dichromate-concentrated sulfuric acid heating method; available nitrogen content is measured by semi-automatic Kjeldahl method; total nitrogen content is measured by the J200 laser spectral elemental analyzer; available phosphorus content is measured by NaHCO$_3$ colorimetric method; available potassium content is measured by the flame photometer method; soil mechanical composition was measured using Mastersizer 2000 laser particle sizer; soil bulk density and soil water content was measured by the classical drying method. Soil erosion intensity was downloaded from the Resource and Environmental Science and Data Center, Chinese Academy of Sciences(RESDC) [34].

### 3.3 Data analysis

**(1) Soil Quality Index (SQI) calculation method.** First, the total data set (TDS) for evaluating the soil quality index was constructed. Based on the comprehensive review of published

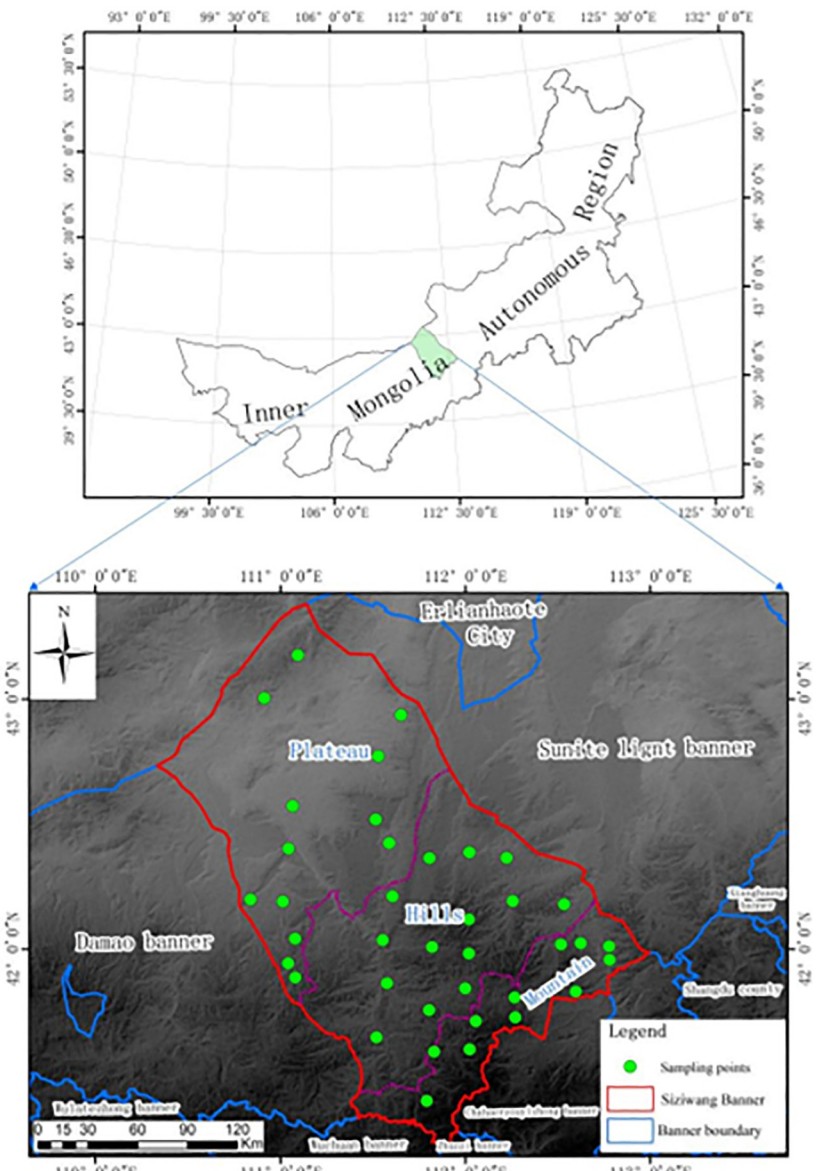

**Fig 1. Location of study area and map of sampling points.**

studies based on the academic searches in the Web of Science Core Collection database and CNKI database using the keywords of "soil quality assessment" and "soil health assessment" [11], a total number of 415 literature was selected from 1990 to 2019, among which, the minimum data set (MDS) was used in 155 literature for soil quality evaluation. These results indicated that the widely used physical indicators, ranked by the using frequency, were soil bulk density, soil water content and soil mechanical composition, etc., while the widely used chemical indicators were organic matter, available phosphorus, total nitrogen, available nitrogen, available potassium, etc., besides, soil pH was the most widely used environmental indicator. Thus, these indicators were used in this study due to their popularity in previous studies. In addition, the soil erosion modulus was also used in this study in consideration of the great impacts of long-term serious soil erosion on soil quality. Finally, 12 indicators, which were soil

bulk density, water content, mechanical composition (clay grain, sand grain, mealy sand grains), organic matter, available potassium, available phosphorus, available nitrogen, total nitrogen, pH and soil erosion modulus, were selected as TDS.

The next step is to construct the MDS. Principal component analysis was applied to reduce the dimensionality of TDS to screen out the major components with the characteristic root values higher than 1. Norm values of each indicator in the principal component factors were also calculated, where a larger Norm value means a large combined loading of the indicator on all principal components, indicating that more soil quality information was included in this indicator. If the Norm value of one indicator falls within 10% of the maximum Norm value, this indicator was selected, and then the correlation analysis was performed to select the indicators with the correlations higher than 0.5 in each component factor, and then screen out the collinearity factors based on the selected indicators. The Norm value was estimated using the formula (1):

$$N_{ik} = \sqrt{\sum_{j=1}^{k} (u_{ik}^2 e_k)} \tag{1}$$

Where, $N_{ik}$ represents the Norm value of the $i$th indicator in the first k principal components with eigenvalues greater than 1; $u_{ik}$ represents the loading of the $i$th indicator in the kth principal component, $e_k$ is the eigenvalue of the kth principal component.

Finally, the SQI of TDS and MDS were calculated, respectively. The higher the SQI value, the better the soil quality. The SQI was estimated as follows:

$$SQI = \sum_{i=1}^{n} w_i s_i \tag{2}$$

Where, $s_i$ represents the score of each indicator, 0~1, $n$ is the number of indicators, and $w_i$ represents the indicator weight value, which is the ratio of the value of the variance of $i$th factor to the sum of the variances of all the common factors by conducting factor analysis on the indicators. Where the formula for calculating $s_i$ is:

$$s_i = \frac{a}{1 + (x/x_0)^b} \tag{3}$$

Where: $a$ is the maximum score, the default value is 1, $x$ is the measured value of soil indicator, $x_0$ is the average value of the soil indicator, and $b$ is the slope of the equation, for the indicators of the "the more the better" type, the $b$ value was quantified as -2.5 and while the value for the indicators of the "the less the better" type was quantified as 2.5 [35]. In this study, the soil water and soil nutrient indicators in the arid zone can facilitate plant growth and development and improve the soil physical properties, and soil fertility, thus these indicators can be quantified as the"the more the better" type [36]. The sand content, silt content of the soil has the low water holding capacity and poor fertility, thus these indicators can be quantified as the "the less the better" type, while the clay content are suitable for the "the more the better" type due to its richness in soil nutrients [36]. The indicator soil erosion represent the soil loss grade, thus is suitable for "the less the better" type; Since the soil in the study area was alkaline, if the pH value get higher, the growth of grass will be limited due to the high alkalinity, thus the indicator pH was quantified as the "the less the better" model. For the indicator soil bulk density, the soil porosity decreased as the density increased, thus it belonged to "the less the better" type. Thus the $b$ value of each indicator is presented in Table 1.

**(2) Soil Quality index reasonableness verification.**   The validation of the rationality of the MDS evaluation index system is important for soil quality evaluation [26] since the

**Table 1. Soil index score model b value.**

| Indicator | *b* value | Indicator | *b* value |
|---|---|---|---|
| soil organic matter | -2.5 | Sand grains | 2.5 |
| available potassium | -2.5 | total nitrogen | -2.5 |
| available phosphorus | -2.5 | pH | 2.5 |
| available nitrogen | -2.5 | soil erosion intensity | 2.5 |
| clay grain | -2.5 | soil bulk density | 2.5 |
| Mealy sand grains | 2.5 | soil water content | -2.5 |

rationality of the selection of the MDS indicators usually has a great impact on the accuracy of soil quality evaluation. In this study, the range, mean value, and the variation coefficients of SQI calculated by the two methods of TDS and MDS were compared to evaluate the variability of the assessment results between the two methods. Meanover, the linear regression analysis and Pearson correlation analysis were conducted for the SQI values based on the TDS and MDS methods to detect their correlations. At the same time, the discrepancy analysis on the distribution characteristics of the indicator values and SQI values was conducted based on the hypothesis of the geography detector model, to further judge the rationality of the selection of the indicators.

## 4 Results

### 4.1 Soil quality index calculation

**(1) Minimum set of indicators and determination of indicator weights.** In this study, four main components were screened with a cumulative contribution of 82.98%. These four components were finally quantified as evaluation indicators of MDS: soil organic matter content, sand content, soil erosion class and pH (Table 2). The results of variance and weights estimated based on TDS and MDS are shown in Table 3.

**Table 2. Results of principal component analysis for the TDS indicators.**

| Index | Group | Component | | | | Norm |
|---|---|---|---|---|---|---|
| | | 1 | 2 | 3 | 4 | |
| SOM | 1 | 0.88 | -0.16 | 0.33 | -0.20 | 1.90 |
| AK | 1 | 0.64 | -0.04 | 0.61 | 0.09 | 1.53 |
| AP | 1 | 0.78 | 0.12 | 0.22 | 0.22 | 1.67 |
| AN | 1 | 0.71 | -0.05 | -0.26 | 0.56 | 1.66 |
| Clay | 2 | 0.25 | 0.83 | -0.04 | 0.14 | 1.45 |
| Silt | 2 | 0.00 | 0.94 | 0.07 | -0.25 | 1.56 |
| Sand | 2 | -0.04 | -0.96 | -0.07 | 0.23 | 1.58 |
| TN | 1 | 0.79 | -0.17 | 0.40 | -0.22 | 1.76 |
| pH | 4 | -0.48 | -0.27 | 0.37 | -0.69 | 1.45 |
| Soil erosion intensity | 3 | -0.38 | 0.07 | 0.49 | 0.22 | 1.03 |
| Soil Bulk density | 1 | -0.63 | 0.11 | 0.47 | 0.52 | 1.57 |
| Soil moisture content | 1 | 0.73 | -0.05 | -0.44 | -0.32 | 1.65 |
| Characteristic value of principal component | | 4.29 | 2.64 | 1.56 | 1.47 | - |
| Contribution rate of principal component variance | | 35.73 | 21.99 | 12.97 | 12.29 | - |
| Cumulative contribution rate of principal components | | 35.73 | 57.72 | 70.69 | 82.98 | - |

**Table 3. Variance and weights estimated from both TDS and MDS indicators.**

| Index | TDS | | MDS | |
|---|---|---|---|---|
| | Common factor variance | Weight | Common factor variance | Weight |
| SOM | 0.946 | 0.10 | 0.512 | 0.21 |
| AK | 0.787 | 0.08 | - | - |
| AP | 0.717 | 0.07 | - | - |
| AN | 0.891 | 0.09 | - | - |
| Clay | 0.769 | 0.08 | - | - |
| Silt | 0.946 | 0.10 | - | - |
| Sand | 0.97 | 0.10 | 0.817 | 0.33 |
| TN | 0.866 | 0.09 | - | - |
| pH | 0.913 | 0.09 | 0.577 | 0.23 |
| Soil erosion intensity | 0.432 | 0.04 | 0.565 | 0.23 |
| Soil Bulk density | 0.894 | 0.09 | - | - |
| Soil moisture content | 0.825 | 0.08 | - | - |

**(2) Minimum data set Soil Quality Index (SQI-MDS) calculation.** Based on the calculation results from formula 2, the values of SQI at 0–30 cm depth mainly falls in the range of 0.4–0.6, among which, the area proportion for the values from 0.4 to 0.5 was approximately 59% while the area proportion for the values from 0.5 to 0.6 was about 38% (Fig 2).

From the vertical perspective, the area proportion of SQI within the range 0.4–0.5 at the 0–5 cm depth was about 62.65% while the area proportion within the range 0.5–0.6 was about 37.35%, indicating that the SQI values slightly fluctuated arond the mean value of 0.5 (Table 4). Compared with the 0–5 cm depth, the area proportion at 5–10 cm depth for the value range 0.4–0.5 was 23% higher than the area proportion in the 0–5 cm depth, while the area proportion at 5–10 cm depth for the value range 0.5–0.6 was 28% lower than the area

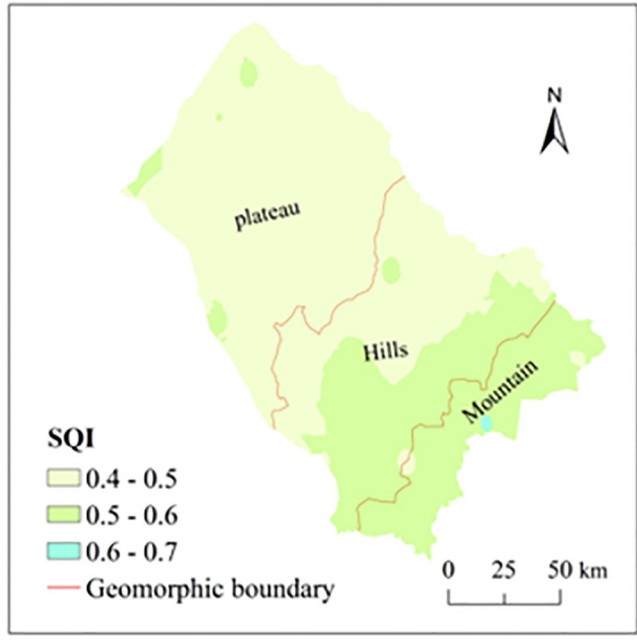

**Fig 2. Distribution of SQI values at 0-30cm depth in the Siziwang Banner.**

**Table 4. Area proportion of SQI values at different depths (%).**

| SQI Grade | | Soil depth (cm) | | | | | | |
|---|---|---|---|---|---|---|---|---|
| | | 0–30 | 0–5 | 5–10 | 10–15 | 15–20 | 20–25 | 25–30 |
| I (High) | SQI≥0.8 | | | | | | | |
| II (Medium-High) | 0.6≤SQI<0.8 | 2.16 | | | 1.9 | | 7.05 | 3.99 |
| III (Medium) | 0.5≤SQI<0.6 | 37.84 | 37.35 | 9.12 | 35.28 | 48.87 | 86.99 | 43.48 |
| | 0.4≤SQI<0.5 | 59.14 | 62.65 | 85.72 | 62.82 | 51.13 | 5.96 | 52.53 |
| IV (Low-Medium) | 0.4>SQI≥0.2 | 0.86 | | 5.16 | | | | |
| V (Low) | SQI<0.2 | | | | | | | |

proportion in the 0–5 cm depth, and the area proportion of SQI <0.4 accounted for about 5%, indicating that, the soil quality at 0–5 cm depth is greater than that at 5–10 cm depth. Generally speaking, the soil quality at 0–30 cm depth generally increases with the increase in the depth. The soil quality in the Siziwang Banner was mainly in the medium level (Table 4).

## 4.2 Validation of the rationality of the SQI-MDS

**(1) Comparison of the SQI-TDS and SQI-MDS.** The weights of TDS indicators are presented in Table 3, and the b-values in the model for calculating the scores of each indicator are presented in Table 1. Based on the calculation, the results indicated that the SQI-TDS falls in the range of 0.32–0.69 with the mean value of 0.49 and variation coefficient (CV) of 18.9%, while the SQI-MDS ranged from 0.38 to 0.73 with the mean value of 0.50 and CV of 15.2%. It can be found that the value range, the mean value and the CV are quite similar between SQI-TDS and SQI-MDS. A good linear relationship between SQI-TDS and SQI-MDS was found with the determination coefficient of 0.497 (Fig 3). The Pearson correlation analysis indicated the correlation coefficient of SQI-TDS and SQI-MDS was 0.706 ($p < 0.001$). It can be concluded that both the MDS method can be effectively reflect the soil quality in the Siziwang Banner, and can be representative as the evaluation based on the whole data set using the

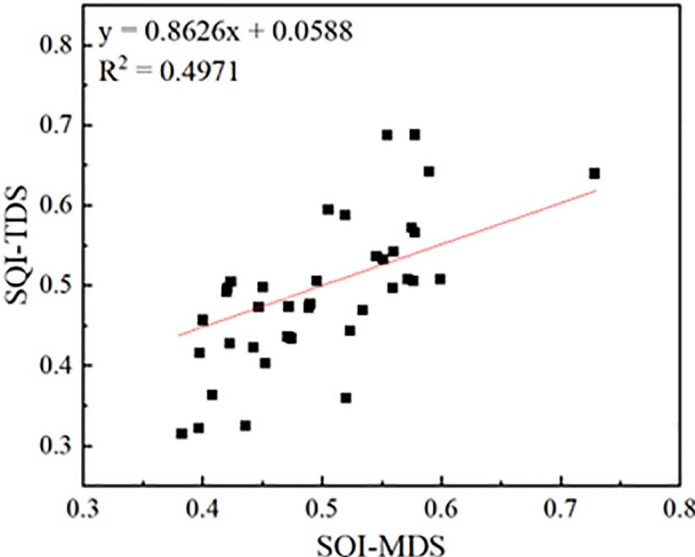

**Fig 3. Linear regression between SQI-MDS and SQI-TDS.**

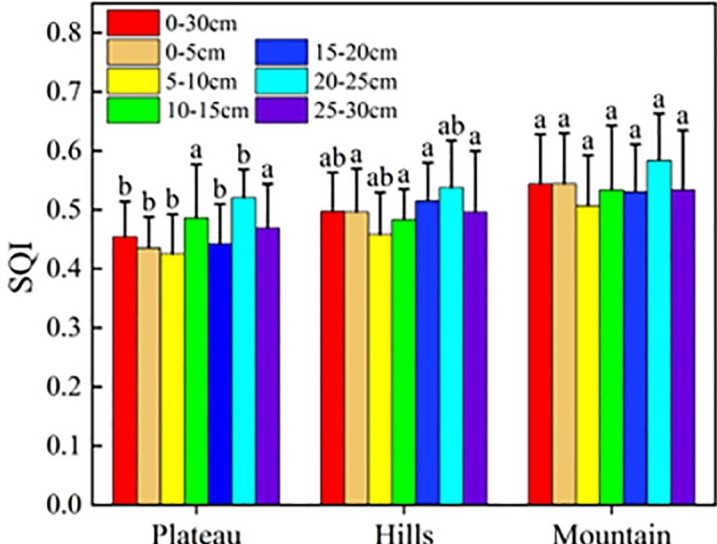

**Fig 4. Soil quality index(SQI) of plateau, hills and mountains in the study area.**

TDS method. The soil quality assessment in the Siziwang Banner can be conducted using the indicators of the soil organic content, sand content, soil erosion grade and pH value.

**(2) Characterization of SQI-MDS and indicator factor distribution.** It can be seen from Fig 4 that the SQI values at all depths were the highest in the south mountains, followed by the central hills, while the values in the north plateau were the lowest. The statistics analysis indicated that significant differences were found between the south mountains and the north plateau at the 0–30 cm depth ($P < 0.05$). The significant differences were also found at the 0–5, 5–10, 15–20 and 20–25 cm depth between the the south mountains and the north plateau (Fig 4).

Similar to the SQI, the soil organic matter also decreased from the south mountain, to the central hill, to the north plateau (Fig 5), and significant differences were reported for the three

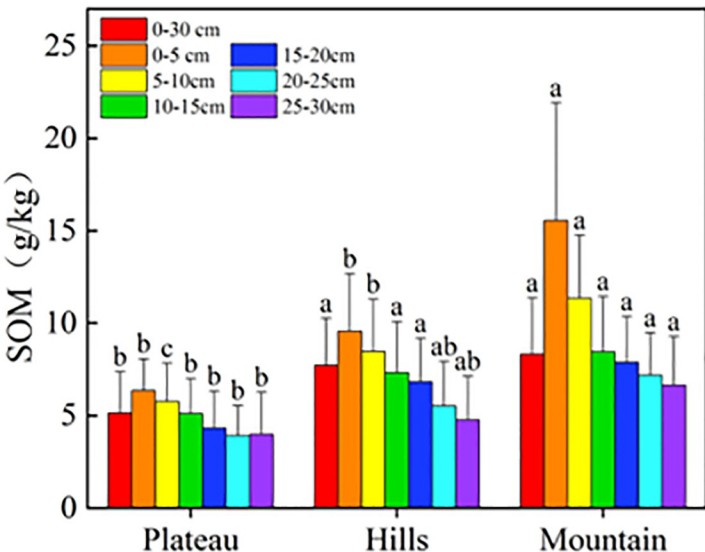

**Fig 5. Characteristics of SOM in the plateau, hills and mountains of the study area.**

regions (P<0.05) at the 0–10 cm depth: the average SOM value in the mountains was 1.5 times higher than in the hills, and 2.5 times higher than on the plateau, indicating the substantial differences in SOM content among the three regions. The SOM content in the Siziwang Banner mainly falls in the range of 11–16.5 g kg$^{-1}$, which was similar with the values of 10.2–14.1 g kg$^{-1}$ reported in the Xilamuren desert steppe [37].

The geographical distribution of soil mechanical composition revealed no obvious differences across the Siziwang Banner. The overall soil sand content accounted for 55–68% among different soil depths, while the clay contents were the lowest for different depths in the range of 1.1–2.9%. Previous studies have indicated that the threshold wind velocity of sand-moving in the desert steppe in the Inner Mongolia was about 6 m s$^{-1}$. Based on the measured data of wind speed from the Siziwang meteorological station from 1990–2020, the threshold wind for sand-moving have been recorded in 7702 days, accounting for 68% of the total days (11296 days) of the period 1990–2020. As known, the Siziwang Banner has suffered from serious soil erosion, based on the study conducted by the Resource and Environmental Science and Data Center, Chinese Academy of Sciences, wind erosion occurs across the entire Siziwang Banner, with the erosion grade in most regions higher than the moderate grade. More specifically, for the 38 sampling sites, 13, 5, 8 and 1 site were in the moderate, strong, very strong and extremely strong erosion of wind erosion, respectively, indicating that wind erosion was very serious in the Siziwang Banner, which has resulted in the coarse sand soil in this region. Previous studies have also indicated that wind erosion could pose great adverse impacts on soil mechanical composition [38–43], thus the sandy soil in the study region can effectively reflect the impacts of soil erosion on soil quality.

In this study, the soil pH value is greater than 7, and the soil is alkaline. The average values in mountainous, hilly and plateau areas are 8.14, 8.17 and 8.43, respectively. The overall pH shows an increasing trend from southern mountainous areas to northern plateau. The high value area of soil pH in this study corresponds to the low value area of SQI, which shows that the stronger the soil alkalinity is, the worse the soil quality is. This result objectively reflects the impact of the soil pH value on the soil quality in the study area (Fig 6).

In the total data set, indicators such as total nitrogen, available phosphorus, available nitrogen, and available potassium also showed similar characteristics with the soil quality index and

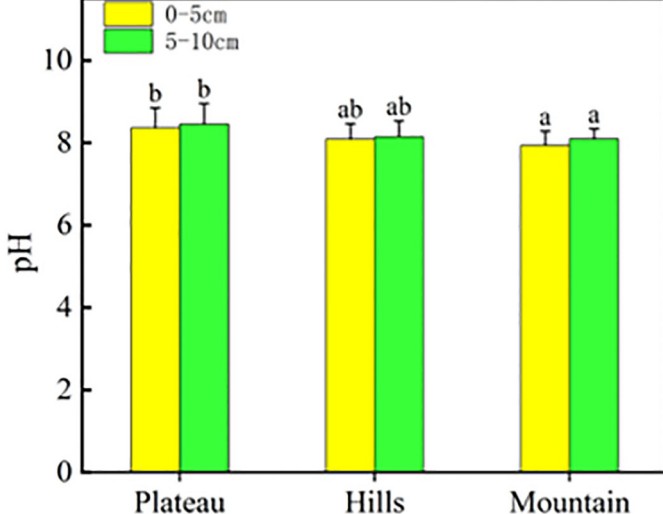

**Fig 6. pH characteristics of the Plateau, Hills and Mountain in the study area.**

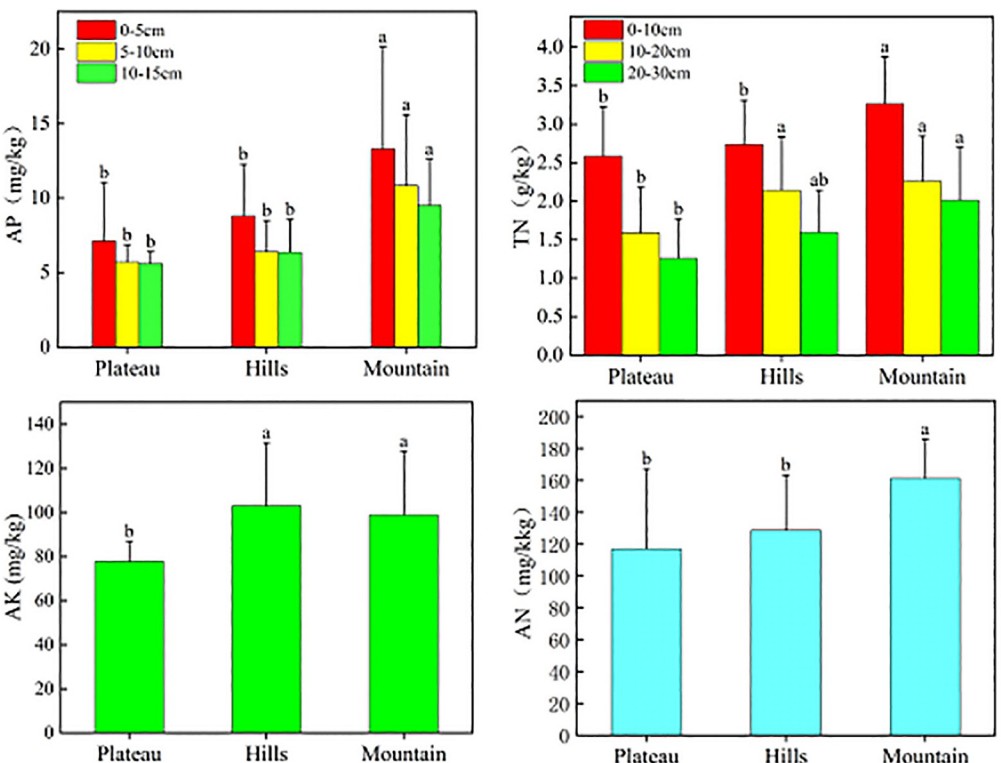

**Fig 7. Characteristics of AP, TN, AK, and AN in the Plateau, Hills, and Mountain in the Study Area.**

soil organic matter at different soil depths, regional differences were also found for these indicators among the south mountain, central hill and the north plateau (Fig 7). In this study, the indicators of soil organic matter, sand content, pH, and soil erosion grade were selected to reflect the soil quality in the study region, which has avoided the cost and time consumption for the field and lab experiments for all the soil indicators, thus has provided an efficiency way for evaluating the soil quality in the desert steppe.

## 5 Discussion

### 5.1 Importance of soil quality assessment in desert steppe

The arid environment of desert steppe in Inner Mongolia, combined with strong wind and frequent wind erosion, has led to the decline of soil quality and productivity. Sustainable productivity can be ensured only through ongoing monitoring of soil quality. Therefore, this article seeks a relatively scientific assessment method to determine the soil quality of the region by constructing a minimum index set for soil quality assessment of the desert steppe. Soil quality results in the region were found to be moderate. The mean SQI value in this study is slightly higher than 0.351 in Ningxia's Yanchi Desert Steppe [44], which was probably due to the relatively low sand quality index in Ningxia due to the high proportion of sandy area in Ningxia. The results of this study are similar to those reported in the Xilingole steppe [45]. Overall, soil quality in the study area was low. In the future, how to reverse poor soil quality may be a priority issue. It is necessary to focus on considering targeted strategies from the minimum dataset indicators. This kind of research idea can lighten the load of soil quality evaluation, and the research method and result can provide reference for monitoring and evaluation of soil quality in surrounding area.

## 5.2 Important indicators affecting soil quality in desert steppe

The main reason why soil organic matter can become one of the important indicators is that as a component of soil solid matter, it forms organic inorganic composite colloids with inorganic colloids in the soil, thereby enhancing soil aggregation and indirectly improving soil erosion resistance. Other scholars have also found similar views. Six J (2002) found that organic matter, as an important cementitious material in soil [46], has strong corrosion resistance when formed in the presence of organic matter. The higher the content of organic matter, the greater the aggregate content in the soil, the stronger the soil's corrosion resistance. Greenland D. J (1961) also found that sugars in organic matter can also cause soil aggregation and enhance soil erosion resistance [47]. Chaofu Wei (1995) and Mingkui Zhang (1996) found that the quantity and stability of water stable aggregates were positively correlated with the content of organic matter [48,49]. It can be seen that the content of soil organic matter, as erodible particles, directly restricts soil wind erosion, thereby affecting regional soil quality.

The important reason why sand particles are used as an important indicator in evaluating soil quality in this study is that although the particles of sand particles are relatively large and are less likely to form aggregate structures compared to silt and clay particles, previous studies have shown that when there are large amounts of sand particles in the soil, the proportion of clay and silt particles decreases correspondingly, and the number of aggregates decreases, which can reduce the stability of the soil structure and make the soil susceptible to wind erosion [50]. Chepil (1955b) also found that the coarsest and finest soils are more susceptible to wind erosion than moderately textured soils [51]. In addition, Skidmore (1982) wind tunnel experiments found that particles with a particle diameter of < 0.84 mm were erodible particles [52]. Zhibao Dong (1998) and Dong Lei (2022) believe that particles with a particle size distribution of 0.075 to 0.4 mm are easily erodible particles [53,54]. In this study, the range of sand particles and above particle sizes includes the range of erodible and easily erodible particles studied above. Therefore, as an important source of erosion material, its loss can lead to the loss of soil nutrients, resulting in a decline in soil quality.

pH can alter soil structure and hydraulic properties, as well as biogeochemical cycles, resulting in serious harm to ecosystems [55–57]. Soils with high pH values are distributed in calcareous, alkaline, saline-alkali, or sodium soils (pH > 8 or higher). Such soils typically have severe nutritional limitations (deficiencies of cations, micronutrients, and P) and/or toxicity of Na, coupled with excessive $HCO_3^-$, water scarcity (such soils often occur in arid areas), mechanical resistance, and poor aeration. Alkalinized soils typically result in limited litter decomposition, nutrient loss Soil health issues such as decreased soil microbial activity [58–60]. Grassland ecosystems are a major component of terrestrial ecosystems, which are experiencing the impact of global climate change and soil pH changes caused by human interference [61,62]. As a key factor for global change affecting terrestrial ecosystems, changes in soil pH values can affect the biogeochemical cycle, species diversity, and productivity sustainability of grassland ecosystems [63–65]. Overall, this indicator should be used as an important indicator for assessing grassland soil quality.

## 5.3 Deficiencies in this study

It is important to note that although different methods, models, and classifications of soil quality classes are used for soil quality studies [66,67], most studies tend to select limiting indicators of soil fertility to assess soil quality [11,68], whereas the soil quality assessment in this study was a large-scale survey sampling and determination of soil quality background factors over a wide area, which is a large workload and thus may have overlooked Grazing intensity, seasonal grazing, rotational grazing and grassland protection measures may have an impact on

soil quality, and these actions may have a processive impact on soil quality but the extent of the impact is still unclear; therefore, to obtain more accurate soil quality assessment results, it may be that not only direct factors of soil quality need to be quantitatively assessed, but possible anthropogenic cause investigations should also be included in future work.

## 6 Conclusions

1. The minimum data set method was used in the steppe desert in the Inner Mongolia to asses the soil quality and provided reliable results compared with the total data set method;

2. The soil quality index in the Siziwang Banner mainly falls in the range of 0.4–0.6, more than 95% of the total area has a medium level soil quality, indicating that much work should be done in the future to improve the soil quality;

3. By indicating that the evaluation results of MDS indicators (soil organic matter, sand, pH, soil erosion intensity) in this study can represent the soil quality of the study area, this study has provided a simple and effective method for evaluating soil quality in the desert steppe.

## Acknowledgments

We would like to thank all the members of the team for their efforts, and we would also like to thank Dr. Qiankun Guo and Dr. Yuming Zhao for their valuable comments during the writing of the paper, and especially Dr. Yousheng Wang for his support to our team during the graphical work.

## Author Contributions

**Data curation:** Xiaojia Li, Chunrong Guo.

**Formal analysis:** Zheru Hu.

**Funding acquisition:** Xiaojia Li, Ruiping Zhou.

**Investigation:** Xiaojia Li, Ruiping Zhou.

**Methodology:** Xiaojia Li.

**Project administration:** Yufeng Zhang.

**Resources:** Hongtao Jiang, Chunxing Hai.

**Software:** Hongtao Jiang, Chunrong Guo.

**Supervision:** Chunxing Hai.

**Visualization:** Yufeng Zhang.

**Writing – original draft:** Hongtao Jiang, Chunrong Guo.

**Writing – review & editing:** Chunrong Guo.

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
