## [Decision Letter · Decision Letter 0]

20 Mar 2023

PONE-D-23-02575Evaluation of soil quality in Inner Mongolia desert steppe - A case study of Siziwang BannerPLOS ONE

Dear Dr. guo,

Thank you for submitting your manuscript to PLOS ONE. After careful consideration, we feel that it has merit but does not fully meet PLOS ONE’s publication criteria as it currently stands. Therefore, we invite you to submit a revised version of the manuscript that addresses the points raised during the review process.

We look forward to receiving your revised manuscript.

Kind regards,

Chun Liu

Academic Editor

PLOS ONE

“This study was funded by the Fundamental Research Funds for the Inner Mongolia Normal University (2022JBQN099), Scientific Research Project of Colleges and Universities in Inner Mongolia Autonomous Region (NJZY20019)， Major Special Projects of Science and Technology Plan of Inner Mongolia Autonomous Region（zdzx2018058）.”

“This study was funded by the Fundamental Research Funds for the Inner Mongolia Normal University (2022JBQN099), Scientific Research Project of Colleges and Universities in Inner Mongolia Autonomous Region (NJZY20019)， Major Special Projects of Science and Technology Plan of Inner Mongolia Autonomous Region（zdzx2018058）.”

“This study was funded by the Fundamental Research Funds for the Inner Mongolia Normal University (2022JBQN099), Scientific Research Project of Colleges and Universities in Inner Mongolia Autonomous Region (NJZY20019)， Major Special Projects of Science and Technology Plan of Inner Mongolia Autonomous Region（zdzx2018058）.”

7. Please include a separate caption for each figure in your manuscript.

8. We note that Figures 1 and 2 in your submission contain [map/satellite] images which may be copyrighted. All PLOS content is published under the Creative Commons Attribution License (CC BY 4.0), which means that the manuscript, images, and Supporting Information files will be freely available online, and any third party is permitted to access, download, copy, distribute, and use these materials in any way, even commercially, with proper attribution. For these reasons, we cannot publish previously copyrighted maps or satellite images created using proprietary data, such as Google software (Google Maps, Street View, and Earth). For more information, see our copyright guidelines: http://journals.plos.org/plosone/s/licenses-and-copyright.

a. You may seek permission from the original copyright holder of Figures 1 and 2 to publish the content specifically under the CC BY 4.0 license. 

Reviewers' comments:

Reviewer's Responses to Questions

**Comments to the Author**

1. Is the manuscript technically sound, and do the data support the conclusions?

Reviewer #1: Yes

Reviewer #2: No

2. Has the statistical analysis been performed appropriately and rigorously? 

Reviewer #1: Yes

Reviewer #2: Yes

3. Have the authors made all data underlying the findings in their manuscript fully available?

Reviewer #1: Yes

Reviewer #2: Yes

4. Is the manuscript presented in an intelligible fashion and written in standard English?

Reviewer #1: Yes

Reviewer #2: Yes

5. Review Comments to the Author

Reviewer #1: The reviewer has no background in the soil quality evaluation field. Hence, all comments of this paper are focused on the writing format.

1. In the Abstract, there are "pH" and "PH". If they are expressing the same term, they should be unified.

2. In the Abstract, the sentence "with the changing trends of SOI;,3)" has 3 problems. "SOI" should be "SQI", "," should be removed, and there should be a space between ";" and "3)".

3. In Section 1, "it is necessary to choose appropriate methods Indicators It is of great ..." missing period symbol and removed capitalization.

4. The heading of Section 1 should be bolded in the same format as Sections 2 and 3.

5. The period symbol of the last sentence in Section 2 should not be a separate line.

6. Symbols representing latitude and longitude should not use the full width, and there should be space before and after the symbol "-".

7. Between paragraphs, some have a blank line, some do not, and the paper should be unified with the template format.

8. The URL should not be placed in the body text directly, it should be placed in the Reference and cited in the body text.

9. The heading of Sections 1, 2, 3, and 4 are left-aligned, but Section 5 is not left-aligned, they should be unified including the heading of Acknowledgements and References.

Reviewer #2: This paper evaluated the soil quality of the Inner Mongolia desert steppe and compared the minimum data set (MDS) method and the total data set (TDS) method, which can be representative of the soil quality of the study area. This study is important for further grassland ecological construction and grassland management to improve the soil quality in desert steppees. However, it would be better if improvements were made in the following aspects:

1.The introduction of the paper needs to be further improved.

2. Please check the sentences in the whole paper. Some sentences are not smooth.

3. The paper mentioned the geography detector model, but I did not see where the model were used and which was the result.

4. Figure 4 - Figure 7 all use spatial distribution, but in fact, we cannot see the spatial distribution Figures. Figure 4 - Figure 7 only show the mean values of the region.

5. It is suggested to separate the results and the discussion.

6. The content of the discussion is too little, and it is suggested to supplement and improve it.

7. The title of the paper is "Evaluation of soil quality in Inner Mongolia desert steppe - A case study of Siziwang Banner", but the content of the paper mainly emphasizes the method of minimum data set. It is suggested to highlight the key points, whether it is the method or the results of the region.

8. In the results and discussion part, some contents of the method are recommended to be placed in the part Materials and methods.

6. PLOS authors have the option to publish the peer review history of their article (what does this mean?). If published, this will include your full peer review and any attached files.

Reviewer #1: No

Reviewer #2: No

---

## [Author Response · Author response to Decision Letter 0]

15 Apr 2023

Requirements:1.Please ensure that your manuscript meets PLOS ONE's style requirements, including those for file naming. 

Responds:We have revised the manuscript in accordance with the journal's requirements to ensure that the manuscript meets PLOS ONE's style requirements, including requirements for file naming.

Requirements:2.Please include your tables as part of your main manuscript and remove the individual files. Please note that supplementary tables (should remain/ be uploaded) as separate "supporting information" files.

Responds:We have added the tables to the manuscript as requested by the journal and removed the tables from the separate file.

Requirements:3.In your Methods section, please provide additional information regarding the permits you obtained for the work. Please ensure you have included the full name of the authority that approved the field site access and, if no permits were required, a brief statement explaining why.

Responds:The research methods we used are conventional methods, such as principal component analysis and correlation analysis, which are commonly used in statistical analysis, and we can be absolutely sure that the methods do not involve work permits.

Requirements:4.Please state what role the funders took in the study. If the funders had no role, please state: "The funders had no role in study design, data collection and analysis, decision to publish, or preparation of the manuscript."If this statement is not correct you must amend it as needed.Please include this amended Role of Funder statement in your cover letter; we will change the online submission form on your behalf.

Responds:Funding: This study was supported by the Key Project of Inner Mongolia Autonomous Region Philosophy and Social Science Planning (Research on Optimization of Industrial Structure Based on Natural Resources and Conditions in Inner Mongolia, 2022NDA219) (Yufeng Zhang), and the funder participated in the program involvement of the project and funded the publication cost of the results. This study was also supported by Major Special Projects of Science and Technology Plan of Inner Mongolia Autonomous Region (ZDZX2018058) (Ruiping Zhou), which funded the field sampling and soil sample testing costs.

Requirements:5.Please remove any funding-related text from the manuscript and let us know how you would like to update your Funding Statement. Currently, your Funding Statement reads as follows:

“This study was funded by the Fundamental Research Funds for the Inner Mongolia Normal University (2022JBQN099), Scientific Research Project of Colleges and Universities in Inner Mongolia Autonomous Region (NJZY20019)， Major Special Projects of Science and Technology Plan of Inner Mongolia Autonomous Region（zdzx2018058）.”

Responds:We have removed any grant-related text from the "Acknowledgments" section of the manuscript and updated our statement about the grant here. Please revise the funding statement to read:This study was supported by the Key Project of Inner Mongolia Autonomous Region Philosophy and Social Science Planning (Research on Optimization of Industrial Structure Based on Natural Resources and Conditions in Inner Mongolia, 2022NDA219) (Yufeng Zhang), and the funder participated in the program involvement of the project and funded the publication cost of the results. This study was also supported by Major Special Projects of Science and Technology Plan of Inner Mongolia Autonomous Region (ZDZX2018058) (Ruiping Zhou), which funded the field sampling and soil sample testing costs.

Requirements:6.We note that you have stated that you will provide repository information for your data at acceptance. Should your manuscript be accepted for publication, we will hold it until you provide the relevant accession numbers or DOIs necessary to access your data. If you wish to make changes to your Data Availability statement, please describe these changes in your cover letter and we will update your Data Availability statement to reflect the information you provide.

Responds:Statement on the content of the study data: The data involved in this study are reflected in the manuscript of the paper and no additional databases are required.

Requirements:7.Please include a separate caption for each figure in your manuscript.

Responds:In accordance with the journal's requirements, we have included the figure captions in the manuscript in the appropriate places.

Requirements:8.We require you to either (1) present written permission from the copyright holder to publish these figures specifically under the CC BY 4.0 license, or (2) remove the figures from your submission:

Responds:The satellite images (Digital Elevation Model DEM) and administrative boundaries used in Figure 1 and Figure 2 of this study are from the Resource and Environment Data Center of the Chinese Academy of Sciences, which are public data, and according to their requirements, the publication of the paper requires the data source to be noted in the cited references, and this study has been cited in the references according to their requirements, and the source part of the data has been fully explained.

The following content is a description of the data sharing and the image 1 and image 2 resources download addresses from the Chinese Academy of Sciences.

https://www.cas.cn/tz/201902/t20190220_4679797.shtml

The above link is a document from the Chinese Academy of Sciences on data sharing, with a specific attachment at the bottom of the page: the name of the attachment is "Measures for Scientific Data Management and Open Sharing (for Trial Implementation)" issued by the Chinese Academy of Sciences. Article 26 in Chapter 6 of the annex clearly states that users of scientific data should abide by the code of academic ethics and indicate the scientific data cited in accordance with the relevant standards when publishing papers, applying for patents, publishing monographs, etc. For more details, please see the attachment.

The following is the download address of the DEM file and administrative division file used in the article: you need to register a member account before downloading, the link is as follows: https://www.resdc.cn/data.aspx?DATAID=284

https://www.resdc.cn/DOI/DOI.aspx?DOIID=122

Responds to the reviewer’s comments:

Reviewer 1:

1.Comment: In the Abstract, there are "pH" and "PH". If they are expressing the same term, they should be unified.

Response: We have made corrections based on the reviewers' comments. The correct expression for PH in the text is pH, and all "PH" in the article has been revised to "pH". Please see lines 37, 142, 148, 180, 181 for the revised text.

2.Comment:In the Abstract, the sentence "with the changing trends of SOI;,3)" has 3 problems. "SOI" should be "SQI", "," should be removed, and there should be a space between ";" and "3)".

Response: I am very sorry, due to an oversight on my part, I wrote "SQI" as "SOI". I have now changed "SOI" to "SQI" and also removed the comma and added a space between ";" and "3)"". Please see line 35 for the revised content.

3.Comment:In Section 1, "it is necessary to choose appropriate methods Indicators It is of great ..." missing period symbol and removed capitalization.

Response: The sentence "It is necessary to choose the appropriate methodological indicator" was followed by our addition of a period, while the capital letters that should not be present were removed. Please see line 85-86 for the revised text.

4.Comment:The heading of Section 1 should be bolded in the same format as Sections 2 and 3.

Response: The title format has been completely revised according to the requirements of the journal.

5.Comment:The period symbol of the last sentence in Section 2 should not be a separate line.

Response: The period has been placed in the correct position

6.Comment: Symbols representing latitude and longitude should not use the full width, and there should be space before and after the symbol "-".

Response: We have used the half angle form for the longitude and latitude symbols, and added spaces before and after the "-",Please see line 92 for details of the revised content.

7. Comment:Between paragraphs, some have a blank line, some do not, and the paper should be unified with the template format.

Response: The paragraph format has been modified to a uniform format according to the requirements of the dissertation template

8. Comment: The URL should not be placed in the body text directly, it should be placed in the Reference and cited in the body text.

Response: The form in which the URL appears directly in the text has been modified to be quoted in the literature,Please see line 479 for details of the revised content.

9. Comment:The heading of Sections 1, 2, 3, and 4 are left-aligned, but Section 5 is not left-aligned, they should be unified including the heading of Acknowledgements and References.

Response: All title formats in the article have been uniformly modified according to the template requirements

Reviewer: 2

1.Comment:The introduction of the paper needs to be further improved.

Response:The article introduction has been further improved according to the review requirements.Please see lines 44-89 for refinements.

2.Comment:Please check the sentences in the whole paper. Some sentences are not smooth.

Response: In order to make the sentences expressed in the article more in line with the requirements of the journal, we got native English speakers to help us refine the sentences throughout the article.

3.Comment:The paper mentioned the geography detector model, but I did not see where the model were used and which was the result.

Response: The reason why "geographical detectors" appear in the paper is because the author wants to express an important idea in the development hypothesis of geographical detectors. If the independent variable has a significant impact on the dependent variable, then there must be significant similarity in spatial distribution between the independent variable and the dependent variable. This article does not actually use geographic detectors.

4.Comment:Figure 4 - Figure 7 all use spatial distribution, but in fact, we cannot see the spatial distribution Figures. Figure 4 - Figure 7 only show the mean values of the region.

Response: The actual meaning we want to express through Figure 4 and Figure 7 is the difference in average values between the north and south regions. Therefore, we modify the name of Figure 4 to "Soil quality index(SQI) of plateau, hills and mountains in the study area", and modify the name of Figure 7 to "Characteristics of AP, TN, AK, and AN in the Plateau, Hills, and Mountain in the Study Area"

5.Comment:It is suggested to separate the results and the discussion.

Response: Based on the reviewers' comments, we have described the study results and discussion sections separately in the paper. The revised results and discussion sections are visible in the article in lines 196-357.

6.Comment: The content of the discussion is too little, and it is suggested to supplement and improve it.

Response: According to the comments of the review, we have supplemented and improved the discussion section,Please see lines 283-350 in the article for refinements.

7. Comment:The title of the paper is "Evaluation of soil quality in Inner Mongolia desert steppe - A case study of Siziwang Banner", but the content of the paper mainly emphasizes the method of minimum data set. It is suggested to highlight the key points, whether it is the method or the results of the region.

Response: Based on the review comments, we focused on improving the evaluation of desert grassland soil quality, especially in the discussion section, where we focused on the possible reasons why the indicators involved in the calculation were adopted.

8. Comment:In the results and discussion part, some contents of the method are recommended to be placed in the part Materials and methods.

Response: Based on the reviewers' comments, we have consolidated the "Methods" related content in the Research Results and Discussion section into "Materials and Methods".

---

## [Decision Letter · Decision Letter 1]

7 Aug 2023

Evaluation of soil quality in Inner Mongolia desert steppe - A case study of Siziwang Banner

PONE-D-23-02575R1

Dear Dr. Guo,

We’re pleased to inform you that your manuscript has been judged scientifically suitable for publication and will be formally accepted for publication once it meets all outstanding technical requirements.

Kind regards,

Chun Liu

Academic Editor

PLOS ONE

Additional Editor Comments (optional):

Reviewers' comments:

Reviewer's Responses to Questions

**Comments to the Author**

1. If the authors have adequately addressed your comments raised in a previous round of review and you feel that this manuscript is now acceptable for publication, you may indicate that here to bypass the “Comments to the Author” section, enter your conflict of interest statement in the “Confidential to Editor” section, and submit your "Accept" recommendation.

Reviewer #3: All comments have been addressed

2. Is the manuscript technically sound, and do the data support the conclusions?

Reviewer #3: Yes

3. Has the statistical analysis been performed appropriately and rigorously? 

Reviewer #3: Yes

4. Have the authors made all data underlying the findings in their manuscript fully available?

Reviewer #3: Yes

5. Is the manuscript presented in an intelligible fashion and written in standard English?

Reviewer #3: Yes

6. Review Comments to the Author

Reviewer #3: The revised manuscript has properly addressd the comments from the reviewers, I suggest acceptance of this paper.

7. PLOS authors have the option to publish the peer review history of their article (what does this mean?). If published, this will include your full peer review and any attached files.

Reviewer #3: **Yes: **Xiang Ge

---

## [Editor Report · Acceptance letter]

11 Aug 2023

PONE-D-23-02575R1 

Evaluation of soil quality in Inner Mongolia desert steppe - A case study of Siziwang Banner 

Dear Dr. Guo:

I'm pleased to inform you that your manuscript has been deemed suitable for publication in PLOS ONE. Congratulations! Your manuscript is now with our production department. 

Kind regards, 

on behalf of

Dr. Chun Liu 

Academic Editor

PLOS ONE